# Identification of Functional Transcriptional Binding Sites within Chicken *Abcg2* Gene Promoter and Screening Its Regulators

**DOI:** 10.3390/genes11020186

**Published:** 2020-02-10

**Authors:** Yujuan Zhang, Jinhu Huang, Xiangxiu Li, Ci Fang, Liping Wang

**Affiliations:** 1MOE Joint International Research Laboratory of Animal Health and Food Safety, College of Veterinary Medicine, Nanjing Agricultural University, Nanjing 210095, China; zhangyujuan907@163.com (Y.Z.); jhuang@njau.edu.cn (J.H.); lixx19940226@163.com (X.L.); 2018107020@njau.edu.cn (C.F.); 2School of Biotechnology, Jiangsu University of Science and Technology, Zhenjiang 212018, China

**Keywords:** chicken *Abcg2*, promoter, regulators, LMH, primary hepatocytes

## Abstract

Breast cancer resistance protein (BCRP), an ATP-binding cassette (ABC) half transporter encoded by the *Abcg2* gene, is reported to influence the pharmacokinetics of substrate drugs during clinical therapy. The aim of this study was to clarify the mechanisms that regulate the transcription of the chicken *Abcg2* gene through cloning and characterization of its promoter region. Results showed that the *Abcg2* gene is transcribed by a TATA-less promoter with several putative Sp1 sites upstream from two putative CpG islands. A luciferase reporter assay conducted both in chicken leghorn male hepatoma (LMH) cells and chicken primary hepatocytes mapped a basal promoter to nucleotides −110 to +30, which is responsible for the constitutive expression of *Abcg2*. The 5′-region upstream of the basal promoter was characterized by both positive and negative regulatory domains. Further, using the cell-based reporter gene assay combined with RT-PCR and drug accumulation analysis, we found that four xenobiotics, daidzein, clotrimazole, ivermectin, and lipopolysaccharide (LPS), influence the expression and function of BCRP through significant regulation of the *Abcg2* gene promoter. Interaction sites with the *Abcg2* gene promoter of these four selected regulators were clarified by progressive deletions and mutation assays. This study shed some light on the regulatory mechanisms involved in chicken *Abcg2* gene expression and the results may have far-reaching significance regarding the usage and development of veterinary drugs.

## 1. Introduction

It is currently recognized that the impact of drug transporters on clinically relevant drug disposition and drug–drug interactions (DDIs) is as significant as that of drug-metabolizing enzymes [1,2]. Breast cancer resistance protein (BCRP/*ABCG2*, encoded by the *Abcg2* gene) is a known member of the ATP-binding cassette (ABC) transporter family that utilizes the hydrolysis of ATP to energize the efflux of a broad range of substrates, including commonly used antimicrobial agents licensed in veterinary medicine [3,4]. The FDA accepted BCRP as a key drug transporter involved in clinically relevant DDIs, adverse drug reactions, and therapeutic failure of drugs due to its localization in organs that are important in drug disposition [5,6]. Unpredictable drug effects due to alterations in BCRP expression and activity have been frequently observed during clinical therapy [7,8,9]. Therefore, it is of great clinical importance to elucidate the molecular mechanisms underlying the regulation of BCRP expression. 

Although post-transcriptional and translational regulation are involved in BCRP regulation [10,11,12], it is well documented that the regulation of BCRP mainly occurs at the transcriptional level [13,14,15]. Bailey et al. found that the human *Abcg2* gene promoter lacks a TATA box, but possesses several putative Sp1 sites downstream from a putative CpG island. Furthermore, they also found positive and negative regulatory domains upstream from the core promoter [16]. Later, Szatmari et al. [17] identified a conserved enhancer region containing three functional peroxisome proliferator-activated receptor (PPAR) response elements, which have positive regulatory effects on the transcription of the *Abcg2* gene. Benoki et al. found a novel constitutive androstane receptor (CAR), a responsive element in the distal promoter of human *Abcg2* that enhances the expression of the reporter gene via CAR binding [18]. Recent molecular and pharmacological studies demonstrated that the transcriptional activity of BCRP is mediated by some transcriptional factors binding to the response elements found in the regulatory regions of the *Abcg2* gene [19,20,21]. For example, Ee et al. demonstrated that estradiol activates the *Abcg2* promoter through the estrogen response element in T47D:A18 cells [22], whereas progesterone receptors (PRs) inactivate the *Abcg2* promoter via PR elements in breast cancer cells [23]. Other data showed that the transcription of *Abcg2* is also modulated by a remarkable diversity of xenobiotics, including many widely used drugs such as imatinib, toremifene, buprenorphine, norbuprenorphine, and R-methadone [24,25,26,27,28]. These studies from humans and rodents provided evidence implicating complex mechanisms for the transcriptional regulation of the *Abcg2* gene as a result of a number of factors, including DNA elements, transcription factors, and compounds. Species-specific variations are also present in regard to regulation [29]. Transporter research has become an integral part of veterinary drug development and has attracted the interest of many researchers [4]. However, at present, little is known about the concrete mechanism by which chicken *Abcg2* expression is regulated. Therefore, based on the pivotal role of BCRP in the dispositions of some veterinary-labeled drugs, the aim of this study was to investigate the core promotor and cis-acting elements to address the mechanisms involved in the regulation of the chicken *Abcg2* gene. The current work aimed to further enrich the current knowledge regarding the factors affecting transporter gene expression, including genetic and pharmacologic regulation of promoter activity and hence the transcription of transporter genes.

## 2. Materials and Methods 

### 2.1. Plasmid and Drugs

The pGL3-basic vector and Renilla luciferase assay vector pRL-CMV were generously provided by Professor Qian. Rifampicin, clotrimazole, lipopolysaccharide (LPS), and mitoxantrone were purchased from Sigma (St. Louis, MO, USA). Berberine and daidzein were obtained from Aladdin (Cambridge, MA, USA). Enrofloxacin, florfenicol, tilmicosin, sulfadiazine, ciprofloxacin, doxycycline, cefuroxime sodium, and ivermectin were kind gifts from China Institute of Veterinary Drug Control (Beijing, China). All other chemicals were of analytical grade and were obtained from local suppliers, unless otherwise mentioned.

### 2.2. Cell Lines and Primary Embryonic Hepatocytes

The chicken leghorn male hepatoma cell line (LMH) was obtained from American type culture collection (ATCC, No. CRL-2117). The cells were grown in roswell park memorial institute (RPMI) medium supplemented with 10% fetal bovine serum, 1% glutamine, and 1% penicillin and maintained at 37 ℃ in 5% CO_2_. 

Chicken primary embryonic hepatocytes were isolated from the livers of 9-day old embryonated eggs according to the method described by previous studies [30,31]. In brief, the livers were removed from the embryonated eggs under sterile conditions. After aseptic mincing into small fragments, the liver tissues were incubated in trypsin solution for 10–15 min at 37 ℃. The hepatocytes were collected via centrifugation (1000 rpm, 5 min) and filtrated through a 150 μm mesh. The obtained hepatocytes were cultured in serum-free M199 medium supplemented with 100 IU/mL penicillin, 100 μg/mL streptomycin, 5 μg/mL transferrin, 2 mM glutamine, 1.75 mM N-2-hydroxyethylpiperazine-N-ethane-sulphonicacid (HEPES), and 1 μg/mL hydrocortisone and maintained at 37 ℃ in 5% CO_2_.

### 2.3. Amplification and Sequence Analysis of the Promoter Region of the Chicken Abcg2 Gene

The promoter region of the chicken *Abcg2* gene was amplified via reverse transcriptase polymerase chain reaction (RT-PCR) using adult chicken liver-derived genomic DNA as a template. Based on the chicken genomic sequence 423767, we designed primers to amplify the promoter region of chicken *Abcg2,* i.e., 5′-GAGTCCAGAGAGGGCCATGAA-3′ (sense primer) and 5′-AAAAGGCCCAGCAGAA-3′ (antisense primer), generating a product of 2411 bp from −1071 to +1339 (relative to the transcription start site). The amplicons were verified by sequencing. The putative transcription factor binding sites of the chicken *Abcg2* gene promoter were then predicted using AliBaba2.1 (accessed on March 2019) [32]. The CpG islands were predicted by MethPrimer (accessed on March 2019) [33].

### 2.4. Construction of Reporter Plasmids

A series of plasmids containing fragments with various promoter region sizes of the chicken *Abcg2* gene were constructed in this study. The deletion fragments of the chicken *Abcg2* promoter (−1071/+30, −509/+30, −224/+30, −110/+30, and −27/+30) were separately cloned into a pGL3-basic reporter plasmid via homologous recombination; the generated plasmids were named pGL3-D1, pGL3-D2, pGL3-D3, pGL3-D4, and pGL3-D5. Mutated vectors of the putative chicken xenobiotic receptor (CXR) (−649/−638), nuclear factor-kappaB (NF-κB) (−395/386), glucocorticoid receptor (GR) (-191/-182), and specificity protein 1 (Sp1) (−39/−30) binding sites were generated from pGL3-D1, pGL3-D2, pGL3-D3, and pGL3-D4 using PCR-based, site-directed mutagenesis and named CXR-mut, NF-κB-mut, GR-mut, and Sp1-mut, respectively. The putative estrogen receptor (ER) (−228/−219) binding site mutated vector was also generated from pGL3-D2 and named ER-mut. All of the constructed plasmids were verified by DNA sequencing. The forward and reverse primers set for the plasmid construction were listed in Table 1.

### 2.5. Luciferase Reporter Assay

The LMH cells and chicken hepatocytes were seeded in 24-well culture plates and cultured overnight until grown to 70%–80% confluence in their respective mediums, as mentioned in Section 2.2. The ExFect^®^ 2000 (Vazyme, China, Nanjing, No. T202) was then used to mediate thte transfections using 0.5 µg of the reporter constructs and 2 ng of Renilla luciferase assay vector pRL-CMV to correct the transfection efficiency. A promoter-null plasmid, pGL3-Basic, was also included in the transfection assay as a negative control. After 12 hours, the transfection medium was replaced with the fresh medium either alone or containing individual concentrations of xenobiotics. Cells were incubated for another 6 h and the luciferase activity was measured using a Dual Luciferase Reporter Assay Kit (Vazyme, China, Nanjing, No. DL-101-01). Xenobiotics, including natural compounds (daidzein and berberine), commonly used drugs (rifampicin, clotrimazole, enrofloxacin, florfenicol, tilmicosin, sulfadiazine, ciprofloxacin, doxycycline, cefuroxime sodium, and ivermectin) and LPS, were used in the transfection assay. The concentrations of the above xenobiotics were selected according to the cytotoxicity assay results, where the concentrations were based on cell viabilities of no less than 90% compared with the control group [34]. The final concentrations of all of the xenobiotics were 30 µM, except ivermectin (10 µM) and LPS (50 µg/mL). All of the drugs were dissolved in dimethyl sulfoxide (DMSO) first and then diluted to the specified concentration with culture medium. The final content of the DMSO in every drug solution was 0.1%. The control cells received the same DMSO concentration (0.1%) as the exposure groups. The results of the reporter assay were expressed as a ratio of Firefly luciferase activity to Renilla luciferase activity.

### 2.6. RT-PCR

The LMH cells and chicken hepatocytes were seeded in 12-well plates and incubated overnight until grown to 70%–80% confluence. The cells were then treated with individual xenobiotics for 6 h. The final concentration of each xenobiotic was 30 µM, except ivermectin (10 µM) and LPS (50 µg/mL). After treatment, the total RNA was isolated using TRIzol Reagent (Invitrogen, Carlsbad, CA, No. 10296010) according to the manufacturer’s instructions and treated with DNase to remove any DNA contamination. RNA was quantified using a photometer (Eppendorf, Hamburg, Germany) at 260/280 nm. The RNA integrity was confirmed via 1% agarose gel electrophoresis and ethidium bromide staining. Accordingly, cDNA was synthesized using a reverse transcription kit (Takara, Toyobo, Japan, No. RR047) according to the manufacturer’s protocol. Chicken *Abcg2* was quantified using SYBR Green Realtime PCR Master Mix (Takara, Toyobo, Japan, No. RR420) with a real-time PCR detection system (Bio-Rad Laboratories, Hercules, CA). Chicken β-actin was chosen to be the internal control. The relative levels of *Abcg2* were analyzed using the 2^−ΔΔCt^ method [35].

### 2.7. Drug Accumulation Assay

The LMH cells were seeded in 24-well plates and incubated overnight until grown to 70%–80% confluence and were then treated with individual concentrations of xenobiotics for 6 hours. The final concentrations of the xenobiotics were the same as above. After treatment, the LMH cells were washed with phosphate buffer saline (PBS) and incubated with 5 µM mitoxantrone (a substrate of BCRP) for 60 min. Later, the cells were trypsinized and washed with PBS three times. The fluorescence signals of the samples were recorded using a fluorescence-activated cell sorter (FACS)Calibur with CellQuest Prosofware under 488 nm excitation for mitoxantrone.

### 2.8. Statistical Analysis

All experiments were performed in triplicate and repeated at least in 3 independent experiments. The data were presented as the mean ± standard deviation (SD) of 3 replications. The mean values between the groups were compared using the two-tailed Student’s *t*-test. *p* < 0.05 was considered to be significant and *p* < 0.01 was considered to be extremely significant.

## 3. Results

### 3.1. Several Transcription Factor Binding Sites and CpG Islands Were Predicted and Analyzed in the Chicken Abcg2 Gene Promotor 

A PCR-based approach was used to obtain 2411 bp spanning from −1071 to +1339 bp in the promotor sequence of the *Abcg2* gene from the chicken genomic DNA (Figure 1). This region lacked the canonical TATA box, however, putative Sp1 sites were present, starting at −548, −516, −499, −479, −382, −344, −210, −201, and −39 bp relative to the transcriptional start site (TSS). Two CpG islands (marked with gray background) were predicted in the chicken *Abcg2* promoter downstream from the TSS (Figure 1), indicating that the methylation level of the CpG islands might influence the transcription of chicken *Abcg2*. Several transcription factors binding sites, including CXR (−649/−638), NF-κB (−395/−386), ER (−228/−219), GR −191/−182), and AP-1 (−24/−15 and −22/−13) were recognized in this region (Figure 1).

### 3.2. The Core Regions of the Chicken Abcg2 Gene Promotor Were Identified Using the Dual Luciferase Reporter Assay System

To study the core region of the chicken *Abcg2* gene promoter, we cloned the complete 1101 bp fragment (ahead of the ATG initiation codon), generated five sequentially truncated luciferase reporter constructs, and transfected them into the LMH cells and chicken primary hepatocytes. The transcription activities of the different *Abcg2* gene promoter constructs relative to the activity of the pGL3-Basic vector were detected both in the LMH cells and the chicken hepatocytes. In the LMH cells (Figure 2), the transfection of the pGL3-D1(−1071/+30) construct resulted in a 12.2-fold induction of luciferase activity compared with the pGL3-Basic vector (*p* < 0.01). Transcriptional activity was lower and only 6.7-fold higher than that of the pGL3-Basic vector when the pGL3-D2 (−509/+30) construct was used, indicating the possibility of positive regulatory element(s) between −1,071 bp and −509 bp. Removing the next 285 bases resulted in an increase in luciferase activity of the pGL3-D3 (−224/+30) construct, which was greater than that seen in construct D2 (a change from 6.7-fold to 8.9-fold), thereby further suggesting the presence of negative regulatory element(s) between bases −509 and −224. The removal of the upstream region (−224/−111) from pGL3-D3 led to a slight decrease in the luciferase activity of construct D4 (−110/+30) (7.3-fold), implicating a weak increase in activity in this region. When the sequence was deleted as far down as −27 bp (pGL3-D5), the luciferase activity fell almost to the same level as pGL3-Basic. Similar trends were also observed when the constructs were transfected into the chicken hepatocytes (Figure 2). The luciferase activity changed by 9.9-, 5.1-, 8.5-, and 7.0-fold compared with the pGL3-Basic vector when pGL3-D1, D2, D3, and D4 were transfected into chicken hepatocytes, respectively. Transfection of the pGL3-D5 construct resulted in similar luciferase activity levels in pGL3-Basic. On the basis of these reporter assay results, we focused on the promoter region between −110 and +30, a critical area for chicken *Abcg2* gene transcription.

### 3.3. Several Regulatory Elements Were Recognized in the Chicken Abcg2 Gene Promoter by Site-Directed Mutagenesis

We employed site-directed mutagenesis, which disturbed the recruitment of the transcription factors, to further determine the contributions of different transcription factor binding sites to the activity of the chicken *Abcg2* promoter. Compared with the wild-type control (pGL3-D1), the mutation in the region of putative CXR binding site caused a significant decrease (*p* < 0.05) in the relative luciferase activity levels to 80% and 48% in the LMH cells and chicken primary hepatocytes, respectively (shown in Figure 3). The mutation in the putative ER binding site reduced the luciferase activity levels to 53% and 56% in the LMH cells and chicken primary hepatocytes, respectively. Similarly, when the GR-mut and Sp1-mut constructs were transfected into LMH cells, the luciferase activity levels significantly reduced to 80% and 74%, and in chicken primary hepatocytes, the activity levels reduced to 71% and 65%. These results suggested that the predicted binding sites were positive regulatory elements in the chicken *Abcg2* promoter. However, if the putative NF-κB binding site was mutated, the relative luciferase activity significantly increased to about 173% in the LMH cells and 151% in the chicken primary hepatocytes compared with the wild-type control (pGL3-D1), indicating that a negative regulatory element exists in the chicken *Abcg2* promoter.

### 3.4. Some Xenobiotics Could Alter the Activity of the Chicken Abcg2 Gene Promoter

A luciferase reporter assay was performed to screen the regulators for the chicken *Abcg2* promoter from the commonly used drugs licensed in veterinary medicine, including clotrimazole, enrofloxacin, ciprofloxacin, florfenicol, tilmicosin, sulfadiazine, cefuroxime sodium, doxycycline, and ivermectin, as well as the natural compounds berberine and daidzein. LPS was also testes. As shown in Figure 4, among the selected drugs and natural compounds, ivermectin was the most effective activator of the chicken *Abcg2* gene promotor, causing 3.2- and 4.9-fold inductions in the LMH cells (Figure 4A) and chicken primary hepatocytes (Figure 4B), respectively. For the other drugs, although the *Abcg2* promoter was also activated, the luciferase activities were only changed by 1.4–1.6-fold in the LMH cells (Figure 4A) and 1.5–2.6-fold in the chicken primary hepatocytes (Figure 4B). In this study, we failed to observe activation of the *Abcg2* promoter by florfenicol and ciprofloxacin. However, LPS significantly decreased (*p* < 0.01) the luciferase activity levels of the *Abcg2* promoter, with the reporter gene constructs being 0.56- and 0.6- fold in the LMH cells and chicken hepatocytes, respectively (Figure 4).

### 3.5. Xenobiotics Changed the Chicken Abcg2 mRNA Levels and BCRP Transport Function Related to Their Activities on the Abcg2 Promoter

Next, we sought to determine whether the xenobiotics, which affected the activity of the *Abcg2* promoter, could regulate the expression of *Abcg2* mRNA in LMH cells and chicken hepatocytes (Figure 5A,B). After the treatment of cells for 6 h, berberine and tilmicosin resulted in significant decreases in *Abcg2* mRNA in both cell types (*p* < 0.01). However, daidzein, clotrimazole, enrofloxacin, sulfadiazine, doxycycline, cefuroxime sodium, and ivermectin caused significant increases in *Abcg2* mRNA (*p* < 0.05). In addition, LPS significantly downregulated *Abcg2* mRNA in both cell types (*p* < 0.01). In summary, the variation trends of *Abcg2* mRNA were in line with the variation trends of the *Abcg2* promoter after all xenobiotic treatments, except for berberine and tilmicosin. Thus, the chicken *Abcg2* promoter was activated or inactivated by a remarkably diverse group of synthetic compounds that correspondingly upregulated or downregulated *Abcg2* gene expression.

A mitoxantrone (a substrate of BCRP) accumulation assay was further carried out in the LMH cells to detect whether the efflux ability of BCRP was changed by the above xenobiotics. The results in Figure 5C demonstrate that berberine, daidzein, clotrimazole, ivermectin, and LPS significantly changed (*p* < 0.05) the accumulation of mitoxantrone in LMH cells. Among them, berberine and LPS significantly increased (*p* < 0.01) mitoxantrone accumulation in LMH cells, showing relative fluorescence intensities of 157% and 121%, respectively, and revealing that they could block the mitoxantrone efflux from the cells. However, daidzein, clotrimazole, and ivermectin significantly decreased (*p* < 0.01) mitoxantrone accumulation, with relative fluorescence intensities of 71%, 75%, and 69%, respectively.

### 3.6. Xenobiotics Regulate Activity Levels of the Chicken Abcg2 Gene Promoter Through Different Response Elements, Proven by Deletion and Site-Directed Mutagenic Analysis

The above results indicated that daidzein, clotrimazole, ivermectin, and LPS may regulate the expression of *Abcg2* mRNA through the chicken *Abcg2* promoter and influence the function of BCRP. Therefore, we further identified the interaction sites with the *Abcg2* gene promoter of these four significant regulators using progressive deletions and the mutation assay.

Figure 6A shows that the transfection of the pGL3-D1 construct containing putative CXR binding site resulted in a significant induction of luciferase activity compared with the pGL3-Basic vector in both cell types (*p* < 0.01). However, the induced function of daidzein disappeared when the pGL3-D2 construct was transfected. Therefore, we assumed that the daidzein may regulate the chicken *Abcg2* gene promoter via the putative CXR binding site. To confirm this hypothesis, we introduced mutations into the CXR binding site, then transfected it into LMH cells and chicken hepatocytes and determined the luciferase activity levels. As shown in Figure 6B, if the CXR binding site was mutated, the luciferase activity significantly decreased from 1.49-fold to 1.12-fold in the LMH cells and 3.15-fold to 1.34-fold in the chicken hepatocytes. These results demonstrated that the putative CXR binding site was involved in the response of the *Abcg2* gene promoter induced by daidzein.

Similar to daidzein, clotrimazole activated the chicken *Abcg2* gene promoter only when the pGL3-D1 construct (*p* < 0.01) was transfected compared with pGL3-D2 construct (*p* > 0.05) in both the LMH cells and chicken hepatocytes (Figure 7A). The CXR binding site mutation significantly decreased the luciferase activity from 1.74-fold to 1.12-fold in the LMH cells (Figure 7B) and 2.57-fold to 1.23-fold in the chicken primary hepatocytes (Figure 7C), indicating that clotrimazole activated the *Abcg2* gene promoter via the putative CXR binding site.

For ivermectin, transfection of all the constructs, including pGL3-D1, pGL3-D2, pGL3-D3, and pGL3-D4, led to *Abcg2* gene promoter activation in both cell types (Figure 8A). However, only the mutation of the Sp1 binding site significantly decreased the luciferase activity from 1.62-fold to 1.13-fold in LMH cells (Figure 8B) and 2.74-fold to 1.41-fold in chicken hepatocytes (Figure 8C), indicating that the putative Sp1 binding site was involved in the response of the *Abcg2* gene promoter induced by ivermectin.

LPS showed a significantly (*p* < 0.01) inhibitory effect on the *Abcg2* gene promoter when the pGL3-D1 and pGL3-D2 constructs were transfected into both cell types (Figure 9A). However, the mutation of the NF-κB binding site significantly increased the luciferase activity from 0.45-fold to 0.89-fold in LMH cells (Figure 9B) and 0.61-fold to 0.94-fold in chicken hepatocytes (Figure 9C), indicating that putative NF-κB, rather than the ER or the CXR binding site, was involved in the response of the *Abcg2* gene promoter induced by LPS.

## 4. Discussion

Previous studies suggested that the expression of BCRP is tightly regulated, mainly at the transcriptional level with species-specific induction [28,29]. Promoters, which are usually located upstream of a gene, play a decisive role in gene transcription. Our research showed that the chicken *Abcg2* gene promoter lacks the canonical TATA box, which is often found within 100 bp upstream of the transcriptional start site, but has multiple putative Sp1 binding sites, similar to the promoters of the human *Abcg2*, *Abcb1*, *Abcc1* and *Abcg1* genes [36,37,38]. The GC-rich region, which is highly conserved in most sequences of the reported ABC transporter genes [39,40,41], was found downstream of the transcriptional start site in the chicken *Abcg2* gene and is similar to the porcine *Abcb1* gene [42]. However, this region was different from that of the human *Abcg2* gene [16]. By further comparing the whole promoter sequences of chicken *Abcg2* with those of human and mouse, we observed very low similarity. We demonstrated using a luciferase reporter assay that the 5′-flanking region from −110 to +30 bp of the chicken *Abcb1* gene significantly influenced promoter activity, which was an inconsistent result with previous data in the human *Abcg2* gene showing that a sequence of 312 bp, directly upstream from the *Abcg2* transcriptional start site, conferred basal promoter activity [16]. Thus, this was the main target of the current study to clarify the functional transcriptional binding sites within the chicken *Abcg2* gene promoter which may differ from those found in humans.

Cis-acting elements, including enhancers and silencers regulate, gene transcription efficiency through complex interplays with specific transacting factors. Detecting multiple cis-acting elements is critical to understanding transcriptional regulatory mechanisms and gene expression patterns. The results of this study demonstrated that four distinct cis-acting elements, including putative CXR, ER, SP1, and NF-κB, were responsible for chicken *Abcg2* gene expression. Handschin et al. first reported that the CXR element containing the conserved sequence AG(G/T) TCA played an essential role in regulating the chicken CYP2H1 gene [43]. We found the same sequence in the chicken *Abcg2* gene, which enhanced chicken *Abcg2* gene expression. Similar to the putative CXR element, an ER element was also identified as an enhancer of the chicken *Abcg2* gene. This result was consistent with the previous data showing that ER elements enhance the expression of human BCRP mRNA [22]. The Sp1 element plays a critical role in the transcriptional regulation of a series of genes lacking classical TATA box [44]. An analysis of this study showed that Sp1 plays a positive regulatory effect on chicken *Abcg2* gene expression. The same results were also acquired in our previous study on the porcine Abcb1 gene promoter [42]. The NF-κB element was also found as a silencer of the chicken *Abcg2* gene due to weakened chicken *Abcg2* gene expression, similar to a study reported by Ogretmen and Safa [45], who found that a protein complex consisting of NF-κB exhibited a negative regulatory role on the Abcb1 gene promoter in MCF-7 cells (human breast adenocarcinoma cell line).

Understanding whether a drug is an inducer or an inhibitor of chicken *Abcg2* is also important for predicting drug–drug interactions in veterinary medicine and may be a strategy regarding the development of new drugs. Utilizing a cell-based assay to detect xenobiotic effects on the chicken *Abcg2* promoter could be a practical approach in the screening of BCRP inducers or inhibitors, as evidenced by similar studies regarding the screening of Klotho and PXR gene regulators [46,47]. The results of this study revealed that most of the tested compounds activated the *Abcg2* gene promotor, except for florfenicol, ciprofloxacin, and LPS. Our previous studies showed that many of the compounds, such as berberine, enrofloxacin, and tilmicosin, are substrates of chicken BCRP [32]. These findings collectively imply that the substrates are not only transported by BCRP, but also regulate BCRP expression, which may result in drug–drug interactions and undesirable side-effects when co-administrated in a clinical setting. Notably, LPS strongly suppressed the transcriptional activity of the chicken *Abcg2* promoter, mRNA expression levels, and the transport function of BCRP, thereby suggesting that the net absorption of BCRP substrate drugs increases when bacterial infection occurs in chicken; although this might be favorable in many cases, it bears the risk of undesirable side-effects of the drug with a small margin of safety [48,49].

In conclusion, we comprehensively described the characterization of the chicken *Abcg2* promotor and identified its transcriptional factor binding sites contributing to the complex transcriptional regulation of *Abcg2* by compounds. Nevertheless, limitations exist in this work, including that the involvement of transcription factors regulating the chicken *Abcg2* gene and mediation of xenobiotic regulation were not investigated. These issues will be addressed in our further studies.

## Figures and Tables

**Figure 1 genes-11-00186-f001:**
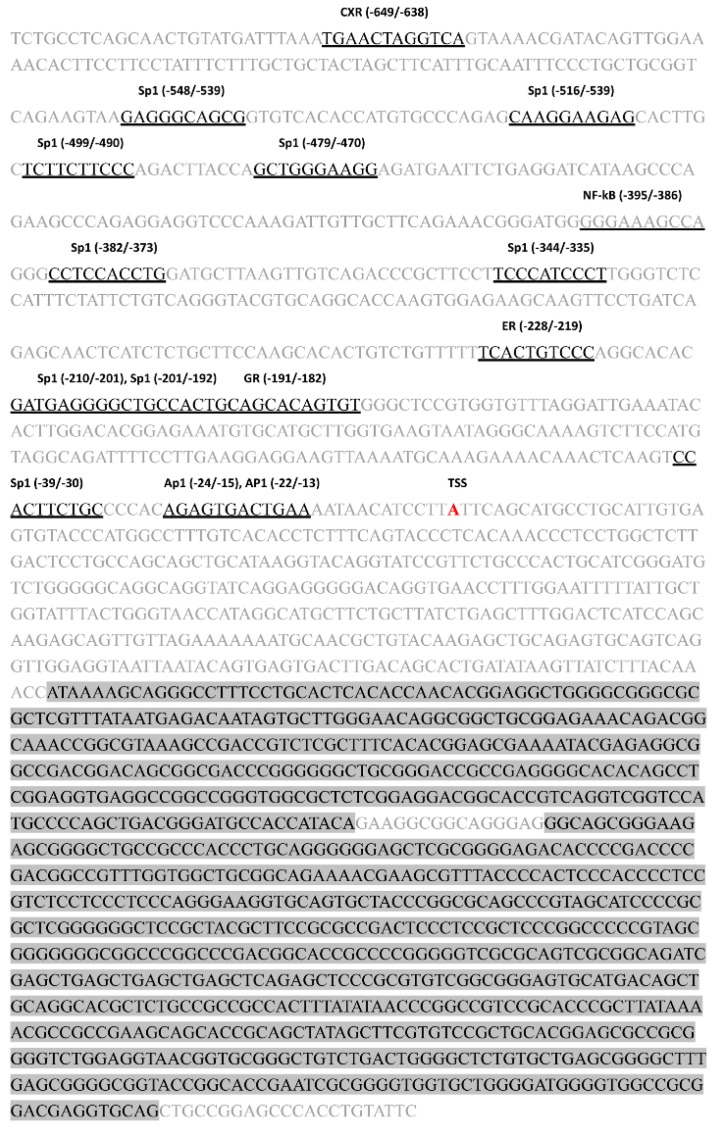
Nucleotide sequence of the promoter region (−1071 to +1339) of the chicken Abcb1 gene. The transcriptional start site (TSS) was designated +1. The putative binding sites for the transcription factors were predicted using the AliBaba2.1 database and marked by underlining that part of the sequence. The CpG islands were predicted by MethPrimer and marked with a gray background.

**Figure 2 genes-11-00186-f002:**
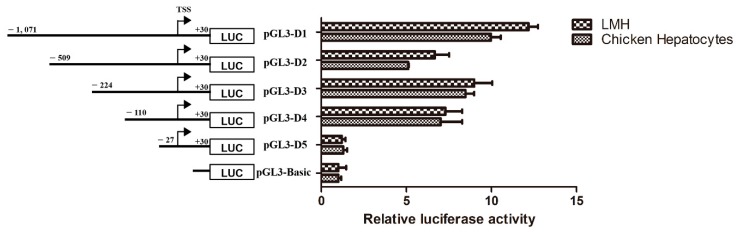
5′-deletion analysis of the chicken *Abcg2* promoter activity. Five promoter constructs were transfected into LMH cells and chicken hepatocytes and assayed for luciferase activity. The left figure shows a schematic diagram of the truncated promoters and the right graph shows the corresponding luciferase reporter assay results. The results are presented as the normalized relative luciferase activity levels (Firefly/Renilla).

**Figure 3 genes-11-00186-f003:**
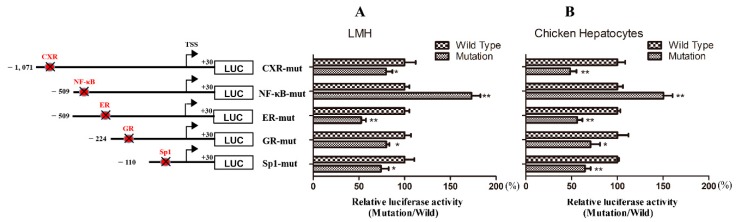
Site-directed mutagenesis analysis of the chicken *Abcg2* promoter activity. Five mutation promoter constructs were transfected into (**A**) LMH cells and (**B**) chicken hepatocytes and assayed for luciferase activity. The left figure shows the schematic diagrams of the truncated promoters and the right graph shows the corresponding luciferase reporter assay results. The results are presented as the normalized relative luciferase activity levels (mutation/wild-type). * represents *p* < 0.05 and ** represents *p* < 0.01.

**Figure 4 genes-11-00186-f004:**
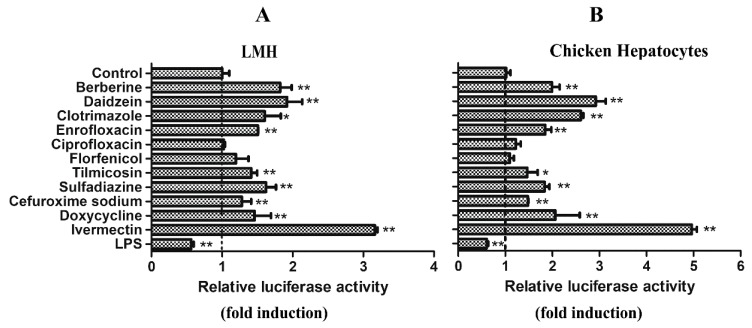
Xenobiotic screening for the chicken *Abcg2* gene promoter. The pGL3-D1 constructs were transiently transfected into (**A**) LMH cells and (**B**) chicken hepatocytes. After 12 hours, the cells were treated with test xenobiotics and the control cells received 0.1% DMSO. Following 6 h of treatment, the dual-luciferase activity levels were determined. Data are shown as the percentages of fold activation of the control cells and represent the means of 3 independent experiments. * represents *p* < 0.05 and ** represents *p* < 0.01.

**Figure 5 genes-11-00186-f005:**
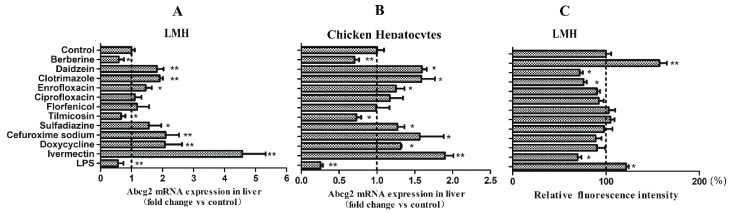
Effects of xenobiotics on the *Abcg2* mRNA levels and breast cancer resistance protein (BCRP) efflux function. (**A**) *Abcg2* mRNA expression in LMH cells; (**B**) *Abcg2* mRNA expression in chicken hepatocytes; (**C**) relative fluorescence intensity of mitoxantrone under 488 nm excitation (xenobiotic treatment/control). * represents *p* < 0.05 and ** represents *p* < 0.01.

**Figure 6 genes-11-00186-f006:**
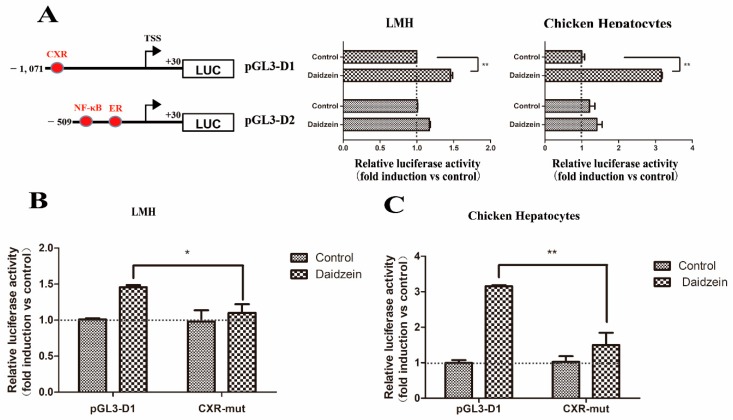
Regulation of chicken *Abcg2* promoter activity by daidzein. (**A**) Isolation of the promoter elements responsible for mediating the stimulation of the *Abcg2* gene promoter activity by daidzein. LMH cells and chicken hepatocytes were transfected with deletion constructs of the *Abcg2* promoter. A schematic illustration of the *Abcg2* gene promoter reporter plasmids used is shown in the left panel. A point mutation was introduced into the pGL3-D1 construct. (**B**) LMH cells and (**C**) chicken hepatocytes were transfected with each mutant. The fold increase in the luciferase activity by daidzein treatment of each construct is indicated. * represents *p* < 0.05 and ** represents *p* < 0.01.

**Figure 7 genes-11-00186-f007:**
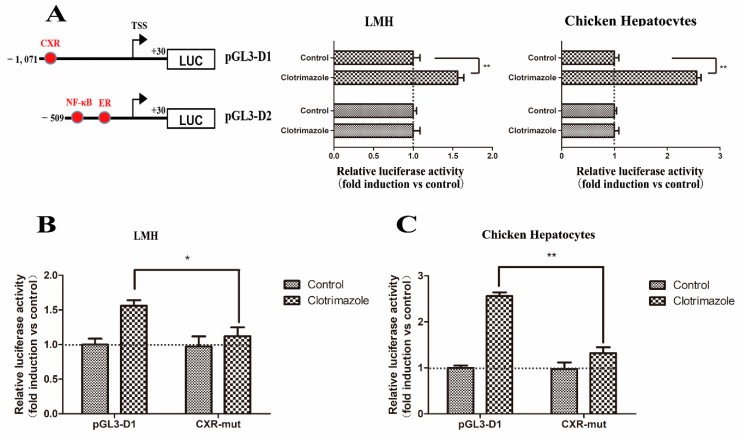
Regulation of chicken *Abcg2* promoter activity by clotrimazole. (**A**) Isolation of the promoter elements responsible for mediating the stimulation of the *Abcg2* gene promoter activity by clotrimazole. LMH cells and chicken hepatocytes were transfected with deletion constructs of the *Abcg2* promoter. A schematic illustration of the *Abcg2* gene promoter reporter plasmids used is shown in the left panel. A point mutation was introduced into the pGL3-D1 construct. (**B**) LMH cells and (**C**) chicken hepatocytes were transfected with each mutant. The fold increase in the luciferase activity by clotrimazole treatment of each construct is indicated. * represents *p* < 0.05 and ** represents *p* < 0.01.

**Figure 8 genes-11-00186-f008:**
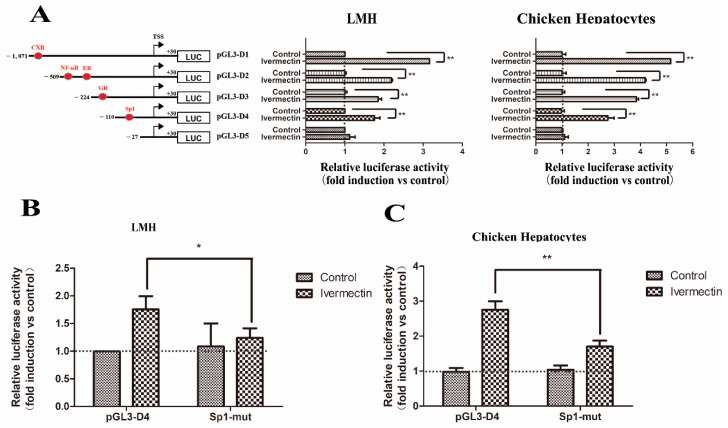
Regulation of chicken *Abcg2* promoter activity by ivermectin. (**A**) Isolation of the promoter elements responsible for mediating the stimulation of the *Abcg2* gene promoter activity by ivermectin. LMH cells and chicken hepatocytes were transfected with deletion constructs of the *Abcg2* promoter. A schematic illustration of the *Abcg2* gene promoter reporter plasmids used is shown in the left panel. A point mutation was introduced into the pGL3-D4 construct. (**B**) LMH cells and (**C**) chicken hepatocytes were transfected with each mutant. The fold increase in the luciferase activity by ivermectin treatment of each construct is indicated. * represents *p* < 0.05 and ** represents *p* < 0.01.

**Figure 9 genes-11-00186-f009:**
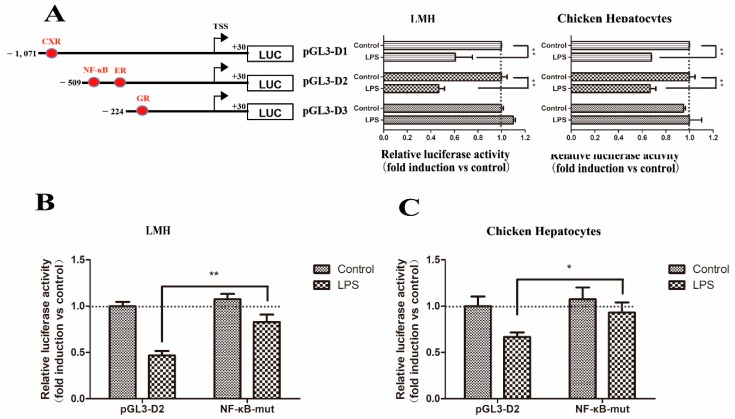
Regulation of chicken *Abcg2* promoter activity by lipopolysaccharide (LPS). (**A**) Isolation of the promoter elements responsible for mediating the stimulation of the *Abcg2* gene promoter activity by LPS. LMH cells and chicken hepatocytes were transfected with deletion constructs of the *Abcg2* promoter. A schematic illustration of the *Abcg2* gene promoter reporter plasmids used is shown in the left panel. A point mutation was introduced into the pGL3-D4 construct. (**B**) LMH cells and (**C**) chicken hepatocytes were transfected with each mutant. The fold increase in the luciferase activity by LPS treatment of each construct is indicated. * represents *p* < 0.05 and ** represents *p* < 0.01.

**Table 1 genes-11-00186-t001:** Primers used in gene cloning and detection.

Name	Primer Sequence (5′–3′)
Primers for plasmid construction	
pGL3-D1-F	atctgcgatctaagtaagcttGAGTCCAGAGAGGGCCATGAA
pGL3-D2-F	atctgcgatctaagtaagcttGAGCACTTGCTCTTCTTCCCAG
pGL3-D3-F	atctgcgatctaagtaagcttTGTCCCAGGCACACGATGA
pGL3-D4-F	atctgcgatctaagtaagcttGCAAAAGTCTTCCATGTAGGCAG
pGL3-D5-F	atctgcgatctaagtaagcttCACAGAGTGACTGAAA
pGL3-deletion-R	cagtaccggaatgccaagcttGGGTACACTCACAATGCAGGC
Primers for Construction of Point Mutation Plasmid	
CXR-mut F	ATTTAAATGCGCACCGACAGTAAAACGATACAGTTGG
CXR-mut R	TTTTACTGTCGGTGCGCATTTAAATCATACAGTTGCT
NF-κB-mut F	AACGGGATGGAGTCACTCCAGGGCCTCCACCTGGATG
NF-κB-mut R	GGCCCTGGAGTGACTCCATCCCGTTTCTGAAGCAA
ER-mut F	GTTTTTTCAGTATACAAGGCACACGATGAGGGGCTGC
ER-mut R	CGTGTGCCTTGTATACTGAAAAAACAGACAGTGTGCT
GR-mut F	TGCCACTGCATAATGGCGTGGGCTCCGTGGTGTTTAG
GR-mut R	GGAGCCCACGCCATTATGCAGTGGCAGCCCCTC
Sp1-mut F	TCCACTTCTGCTACGCATAGTGACTGAAAATAACAT
Sp1-mut R	AGTCACTATGCGTAGCAGAAGTGGACTTGAGTTTGT
Primers for RTP–CR	
*Abcg2* (BCRP)-F	CCTACTTCCTGGCCTTGATGT
*Abcg2* (BCRP)-R	TCGGCCTGCTATAGCTTGAAATC
β-actin F	TGCGTGACATCAAGGAGAAG
β-actin R	TGCCAGGGTACATTGTGGTA

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
