# Peer review of "Identification of Functional Transcriptional Binding Sites within Chicken Abcg2 Gene Promoter and Screening Its Regulators"

_genes, 2020, doi:10.3390/genes11020186_

Round 1
Reviewer 1 Report
The manuscript is related to regulation of the promoter activity and thus the transcription of ABCG2, a member of the group of ABC transporters. The study is interesting due to relevance of BCRP, the product of ABCG2 gene, for pharmacokinetics of many veterinary drugs. The manuscript requires a major revision and the remarks are listed below:
Introduction:
Line 32: “… drug-drug interactions (DDIs) is equal to that of drug-metabolizing enzymes” - Please, change “is equal to” to “as significant as” or “as important as”.
Line 36: “its substantial and strategic localization” is not very suitable expression. ABCG2 cannot make strategy, therefore it can be revised to “its localization”
Material and methods
Page 2, line 76: Please, provide the ATCC number of the cell line.
Page 3: Culture medium was different for the embryonic hepatocytes and hepatoma cell line. Please, explain the reason for that. Hydrocortisone was included in the culture medium for embryonic hepatocytes. This compound can change the expression of ABCG2.
Page 3, Line 108: Table 1 is missing. It should be added.
Page 3, Line 110: The culture medium for both cell lines has not been specified. Please, add this information.
Page 3, Lines 124-125: Please, explain how the concentrations of the drug have been selected?
Results
The data on the figures were presented as mean ± SEM. It is better to use SD.
Pafe 7, Lines 223-225: The drugs were defined as antimicrobial agents. Not all of them belong to antimicrobial agents. Clotrimazole is an antifungal agent. Ivermectin is antiparasitic drug. Florfenicol was used in the study but it was not mentioned among other antibiotics.
“Nature compounds” should be changed to “natural compounds”
Lines 245 and 248: “both cells” is not suitable. It should be changed to “both cell types”
Page 8, Lines 252-253: the sentence is not clear and requires a revision of English
Discussion
Page 11, Line 338: Mdr1 should be changed to ABCB1
Mrp1 should be changed to ABCC1
Line 378: Please, add a reference when function of BCRP is discussed.
Author Response
Dear reviewers,
thank you for the time taken to review the manuscript and for the helpful comments, which will contribute to its overall quality of the manuscript. we have substantially made changes to the manuscript and with anticipation will be considered, as detailed in the attachment. We hope that our changes are reasonable.
Best regards,
Liping Wang

Reviewer 2 Report
Comments for the authors:
Zhang and colleagues present a detailed study of regulatory activity in the chicken Abcg2 promoter and show how xenobiotics contribute to its regulation at several different levels. The experimental design is nicely simple and straightforward, yet yielding interesting and comprehensive results.
I find the study in its present form well presented and complete and have mostly minor comments.
Some sentences in Abstract and Introduction are taken almost verbatim from other sources, such as the sentences in lines 13, 42–44, 44–46 from their respective source papers.
Some basic information on the gene/protein in question is lacking in the introduction. The results shown in Figure 5 suggest that the protein is involved in exporting substrates from cells. Is that correct? Is it known? What is known about the substrates of the transporter? Providing this information would give the reader a better basis to understand the purpose of the study in general and the experiment and the results underlying Figure 5 in particular.
Line 95; I cannot find a program called Meth Primer 3. I found MethPrimer and MethPrimer2, but not Meth Primer 3. A source (link or citation) should be provided.
L. 134; I cannot find a “reverse transcription kit” by Promega. There are plenty of reverse transcription kits, but none with that exact name. To avoid confusion, product numbers should be provided for all materials used.
Line 145; It should be noted at which wavelength measurement takes place, not the name of the channel.
Lines 152–162; Everything that is described here has already been described in Bailey-Dell et al. 2001. Why is this analysis repeated and reported as a new result? (The paper is cited, which is great, but it still is no new result). This could just as well be described in the Introduction instead, because it was already known.
Lines 189, 345; Given that pGL3-D5 has no higher activity than the empty vector (pGL3-Basic), I would argue that the region “critical for chicken Abcg2 gene transcription” should be between -110 and -27, not -110 and +30.
Figure 3, label and legend; “Wildtype” should be spelled out. The samples are not actually wild, they are of wildtype.
Figures 4–9; It would be helpful if there was a line at the reference point, to which the data is normalized (at 1 or 100%). For the same reason, 1 should always be indicated on the axis when data is normalized to this value.
Line 236; This legend states that control cells received 0.1% DMSO. How is that appropriate? That seems to be a different condition than a control condition. Also, this fact should be mentioned in the Methods section and not only in a figure legend.
Line 265; This legend could be more elaborate for panel C. It shows the relative fluorescence intensity of WHAT? What is the tested wave length and what molecule is assessed?
Line 280; After mutational analysis confirmed the functionality of the CXR binding site, I find it odd to still refer to the binding site as “putative.” At this point it has been shown that the binding site is indeed functional, which should be expressed in an adapted language. Similar for lines 295, 308 and 322.
Lines 305–309; What about the activity difference in the other promoter deletion fragments? It seems they all also contain Sp1 sites. Do the authors think that those other fragments also contribute to regulation via their respective Sp1 sites? Could that be tested, at least in another example?
Lines 319–323; What about the ER and CXR sites in that region. Given that they are not mentioned I assume that they did not have an effect, but this should be mentioned explicitly.
Lines 339–340; “indicating that multiple Sp1 binding sites are a common feature of the promoters lacking a TATA box.” This statement seems to be a bit too general. There are a lot more TATA-less promoters than the four mentioned here and multiple Sp1 binding sites can also be found in classical TATA-containing promoters. Is there any quantitative study supporting this statement? Otherwise it should be redacted.
Line 373; This is the first time ceftiofur is mentioned. Do the authors mean ciprofloxacin? Where does this drug name come from?
The English used in the text is not perfect, but certainly understandable. However, there are some places where language errors lead to actual mistakes and such instances that I found are listed here:
- Line 176; This should read “Transcriptional activity was lower and only 6.7 fold higher than that of the pGL3-Basic vector.”
- Line 203; This is not a decrease BY 80% and 48% (which would mean a resulting activity of 20% and 52%), but a decrease TO 80% and 48%.
- Line 210; As above, this should be “increase TO about 173%”
Additional, minor corrections:
- Line 16; To clarify, change to “Previous results showed ...”
- Line 62; Replace “will” with “is to”
- Line 208; “predicated” should be “predicted”
- Line 224 (2x); Those are “natural compounds” but not “nature compounds”
- Line 280; “involved in” should be “is involved in”
- Line 294; “activate” should be “activates”
- Line 309 and 323; “involving in” should be “is involved in”
- Line 373; “showed” should be “shown”
- Line 377; “co-administration in clinic” should be “co-administered in the clinic”
Typos and other minor things:
- Spell out “LPS” once.
- Lines 183, 357, 358; the word “enhancer“ nowadays describes a distant, cis-regulatory element, and this word or derivatives should not be used in the context of regulatory elements in promoters. “Activate”, “increase” or similar words should be used instead.
Figures 6–9; “CON” (“Control”) could be spelled out here. The complete word is shorter than any respective other sample descriptor.
Best wishes!
Author Response
Dear reviewers,
thanks you for the time taken to review the manuscript and for the helpful comments, which will contribute to its overall quality of the manuscript. we have substantially made changes to the manuscript and with anticipation will be considered, as detailed in the attachment. We hope that our changes are reasonable.
Best regards,
Liping Wang
